# Aging and the wandering brain: Age-related differences in the neural correlates of stimulus-independent thoughts

David Maillet[1]*, Roger E. Beaty[2], Areeba Adnan[3], Kieran C. R. Fox[4,5], Gary R. Turner[3], R. Nathan Spreng[6,7]

1 Rotman Research Institute, Baycrest Health Sciences, University of Toronto, North York, ON, Canada, 2 Department of Psychology, Pennsylvania State University, University Park, PA, United States of America, 3 Department of Psychology, York University, Sherman Health Science Research Centre, Keele Campus, Toronto, Canada, 4 Department of Neurology and Neurological Sciences, Stanford University, Stanford, CA, United States of America, 5 School of Medicine, Stanford University, Stanford, CA, United States of America, 6 Laboratory of Brain and Cognition, Montreal Neurological Institute, Department of Neurology and Neurosurgery, Montreal Neurological Institute, McGill University, Montreal, QC, Canada, 7 Departments of Psychology and Psychiatry, McGill University, Montreal, QC, Canada

* davidmaillet@gmail.com

**Data Availability Statement:** The behavioural data for this study can be accessed at: https://osf.io/snmdv/.

## Abstract

In recent years, several studies have indicated that healthy older adults exhibit a reduction in task-unrelated thoughts compared to young adults. However, much less is known regarding age-related differences in time spent engaging in stimulus-independent thoughts or in their neural correlates in the absence of an ongoing task. In the current study, we collected functional magnetic resonance imaging (fMRI) data while 29 young (mean age = 22y) and 22 older (mean age = 70y) adults underwent experience sampling in the absence of an ongoing task (i.e., at "rest"). Although both age groups reported spending a similar amount of time engaged in stimulus-independent thoughts, older adults rated their thoughts as more present-oriented (rather than atemporal) and more novel. Moreover, controlling for these age-related differences in content, we found that experiencing stimulus-independent thoughts was associated with increased posterior cingulate and left angular gyrus activation across age groups compared to exhibiting an external focus of attention. When experiencing stimulus-independent thoughts, younger adults engaged medial and left lateral prefrontal cortex as well as left superior temporal gyrus to a greater degree than older adults. Taken together, our results suggest that, in the absence of an ongoing task, although young and older adults spend a similar amount of time engaging in stimulus-independent thoughts, the content and neural correlates of these thoughts differ with age.

## Introduction

In recent years, many studies have consistently reported that healthy older adults exhibit a reduction in task-unrelated thoughts (commonly referred to as mind-wandering) compared to young adults [for reviews, see 1, 2]. These age-related differences have been found in

**Funding:** This research was supported by grant RFP-15-12 from the Imagination Institute (www.imagination-institute.org), funded by the John Templeton Foundation and awarded to R. B. The opinions expressed in this publication are those of the authors and do not necessarily reflect the view of the Imagination Institute or the John Templeton Foundation. DM was funded through a Canadian Institute of Health Research (http://www.cihr-irsc.gc.ca/e/193.html) postdoctoral training Award.

**Competing interests:** The authors have declared that no competing interests exist.

numerous tasks including those assessing episodic memory encoding [3], sustained attention [4, 5], reading comprehension [6] and working memory [7]. Several explanations have been proposed to account for age-related reductions in task-unrelated thoughts. A first class of explanation centers on the idea that older adults are less able than young to produce task-unrelated thoughts [e.g., 4, 8]. For instance, during the performance of cognitive tasks, older adults may spend more of their cognitive resources on the ongoing task, resulting in fewer resources being available for task-unrelated thoughts. A second class of explanation centers on motivation and affective factors. For instance, older adults are typically more motivated and interested in cognitive tasks, which could make them less likely to exhibit task-unrelated thoughts [6]. Older adults may also have fewer current concerns [9] and exhibit more positive affect compared to young adults [10, 11] both of which may contribute to lower levels of task-unrelated thoughts.

Recently, studies have started to assess age-related differences in the neural correlates of task-unrelated thoughts. In young adults, task-unrelated thoughts are associated with increased activation in several regions including those in default-mode network and fronto-parietal control network [12, 13]. It has been proposed that regions of default-mode network, and particularly the hippocampus, may be involved in the generation of stimulus-independent thoughts [13–15]. Another commonly activated area in this network is the temporopolar cortex [13] which may be involved in semantic processing, theory of mind and/or emotional processing [13, 16]. The frontoparietal network may be involved in the regulation or selection of different streams of thought [13, 15, 16].

One of the first studies to assess age-related differences in the neural correlates of task-unrelated thoughts did so during an episodic encoding task [17]. This relatively challenging task led to very low rates of thoughts completely detached from the ongoing environment, and instead promoted higher numbers of task-related interferences (thoughts about one's performance on the task) and thoughts about the scanner environment. Young and older adults exhibited a similar number of such thoughts, and when compared to on-task thoughts, they were associated with increased activation in precuneus, dorsolateral and lateral temporal cortex in both age groups. Rather than measuring task-unrelated thoughts during fMRI, other studies have related intrinsic brain connectivity to rates of task-unrelated thoughts measured outside the scanner. For instance, one study [18] that included only healthy older adults (but no young adults) found that propensity of having stimulus-independent thoughts was associated with higher intrinsic connectivity between lateral temporal cortex, temporal pole and parahippocampal gyrus and with decreased connectivity between temporal regions and both prefrontal cortex and posterior cingulate cortex. The authors suggested that this pattern of connectivity may promote the generation of thoughts that are shielded from external inputs. In another study [16], age-related differences in task-unrelated thoughts were measured in two tasks varying in difficulty, and patterns of thought were related to resting-state connectivity. Only young, but not older, adults spent more time engaged in task-unrelated thoughts in the easier versus the harder task. Moreover, this modulation of thoughts across tasks was associated with reduced connectivity between left anterior temporal lobe and both medial and left ventrolateral prefrontal cortex in young adults only. The authors suggested that older adults may be unable to flexibly up-regulate patterns of ongoing thoughts when task demands decrease.

The goal of the current study was to assess age-related differences in time spent engaged in stimulus-independent thoughts, which we define here as thoughts that are unrelated to immediate sensory input, as well as their content and neural correlates in the absence of an ongoing task. Specifically, the paradigm alternated between periods of "rest" in which there was no explicit task and thought probes in which participants were asked about their focus of attention in the directly preceding moment. Participants were asked whether they had an external

focus of attention (focusing on visual or auditory perceptions, such as the scanner noise, or on bodily sensations such as being uncomfortable) or whether they were having a stimulus-independent thought. This paradigm was of interested for several reasons. First, many of the factors that have been used to explain age-related reductions in task-unrelated thoughts, such as reductions in cognitive resources, and increased motivation to perform ongoing tasks should be minimized in the absence of a task. These accounts predict that young and older adults should spend a similar amount of time engaged in stimulus-independent thoughts in the absence on an ongoing task, a prediction that we tested here. Note that there is one prior study that measured age-related differences in time spent engaging in stimulus-independent thoughts at rest and found no age-related difference [19]. However, this study used a self-report, retrospective questionnaire which relies on episodic recall, a capacity known to decline with age. Here we assess age-related differences in stimulus-independent thought using an 'in-the-moment', experience sampling method that does not rely on episodic memory but rather probes thoughts, emotions, and experiences as they occur.

Second, the experience of letting one's thoughts wander in the absence of any external task occurs frequently in daily life (e.g. while sitting on a bus, or on a chair by the lake) but we know next to nothing about differences in what young and older adults may be thinking about during these times. In the current experiment, we were interested in age-related differences in several characteristics of stimulus-independent thoughts including temporality, novelty, goal-relatedness, and person-centeredness (self, other, both, or none). We chose these characteristics because although much prior aging research has focused on the detrimental aspects of mind-wandering, such as its link to inattention [7, 17] and negative mood [10, 11], there is reason to believe that these thoughts may also confer certain benefits. In particular, whereas there now exists considerable evidence of a link between mind-wandering and creativity in young adults [for a review, see 20], we do not know of any prior research that has assessed the creativity/novelty of thoughts in older adults. There is also evidence that a large proportion on stimulus-independent thoughts in young adults are social [21] and goal-directed [22, 23] in nature, yet relatively little data is available in older adults.

A characteristic of thought that has often been studied in the aging and mind-wandering literature and that we also measured here is temporality of thought. Whereas some studies have failed to observe age-related differences in temporality of thought [8, 11], another found that young adults reported more future-oriented mind-wandering compared to older adults who instead reported more past-oriented mind-wandering [24]. It was suggested that young adults might engage in more simulation of future outcomes, planning, and decision-making compared to older adults, who might instead interpret, make sense of, and possibly derive satisfaction from remembrance of the past. In that study, older adults also exhibited fewer self-related thoughts compared to young, which the authors suggested could help protect older adults from negative mood and contribute to the age-related positivity bias [25]. Thus, in the current study, one possibility is that young adults would report more future-oriented and self-related thoughts compared to old, indicative of a greater prospective bias in which young adults plan for future outcomes involving themselves. Because future-oriented thoughts tend to be more novel and goal-oriented [26], young adults might also exhibit more of these types of thoughts compared to older adults.

Third, a key strength of our paradigm is that it allows us to assess age-related differences in the neural correlates of in-the-moment stimulus-independent thoughts. The three aforementioned studies relating neural measures with patterns of ongoing thought with age ([16–18]) have all indicated that the lateral temporal cortex and prefrontal cortex are particularly important for stimulus-independent thoughts. In particular, as described earlier, O'Callaghan et al. and Martinon et al. found that age-related differences in connectivity between these regions were associated with reductions in time spent engaging in stimulus-independent thoughts

with age. Based on these prior reports, in the current study, we expected that healthy older adults may exhibit reduced recruitment of lateral temporal cortex and prefrontal cortex during stimulus-independent thought.

## Methods

### Participants

Thirty-two young and 28 older adults were recruited for the study. Young adults were recruited via flyers posted around the University of North Carolina at Greensboro (UNCG) campus as part of a larger project examining individual differences in mind-wandering and creativity. Older adults were recruited from a database of participants who completed previous laboratory studies and expressed interest in participating in future studies. All participants were paid for their time. Two older adults were excluded due to computer errors resulting in missing data. An extra older adult was excluded due to a brain abnormality. An extra three young and three older adults were excluded due to having too few events (<10) in one of the two conditions of interest. This left a total of 29 young (18 females, mean age = 21.60 (SD = 3.65) and 22 older adults (12 females, mean age = 69.90 (SD = 3.50) included in the main analyses presented in this manuscript. Two older adults only contributed one fMRI run of data whereas all other participants completed two runs (see methods section). There were no differences in gender proportions between age groups, $\chi^2$(1, N = 51) = 0.49, p = 0.49. All participants were right-handed with normal or corrected-to- normal vision and no reported history of central nervous system affecting drugs or neurological disease. Older participants had a mean score of 29.6 (SD = 0.91) on the Mini-Mental State Exam (MMSE; [27]) and all scored 27 or above. Participants provided written informed consent. The study was performed in accordance with the guidelines and regulations of University of North Carolina Greensboro Institutional Review Board, who approved the study methods.

### Procedure

In a single scanning session, participants completed two fMRI runs that alternated between periods of "rest", in which there was no explicit task, and thought probes, that asked participants about the content of their thoughts in the directly preceding moment. Each run contained 48 thought probes (total of 96 overall). The thought probes appeared after 8, 10, 12, 14, 16, or 18 seconds with equal probability throughout the experiment and in a random order. Answers to all thought probe questions were self-paced. Thus, duration of each scanning run differed for every subject depending on how long they took to answer the probe questions. On average, each run lasted 15.5 minutes (SD = 1.98) in young adults and 17.6 minutes (SD = 3.38) in older adults, $t$(1,49) = 2.75, $p < 0.01$.

The order of questions asked in each thought probe was fixed. The first question asked participants whether their thoughts were directed towards the external environment or to stimulus-independent thought (1 = completely focused externally, 2 = somewhat externally focused, 3 = somewhat focused on my thoughts, 4 = completely focused on my thoughts). External focus of attention included a focus on visual, auditory and body sensations whereas internal focus referred to thoughts, memories, plans for the future and daydreams (see appendix for full instructions given to participants). If participants chose options 1 or 2 (external focus), the thought probe ended, and rest resumed. If they chose options 3 or 4, additional questions regarding the content of their thoughts were presented. They were asked about the temporal orientation of their thoughts (1 = past, 2 = present, 3 = future, 4 = atemporal), whether the thoughts were related to their goals (1 = related, 2 = unrelated), who the thoughts were about (1 = self only, 2 = others only, 3 = self and others, 4 = none) and about how similar the

thoughts were to those they normally experience (1 = very similar, 2 = somewhat similar, 3 = somewhat different, 4 = very different).

## Behavioral data analysis

For stimulus-independence, we calculated a mean score (1 = completely focused externally, 2 = somewhat externally focused, 3 = somewhat focused on my thoughts, 4 = completely focused on my thoughts). For temporality and person-centeredness, we calculated percentage of each response type. For novelty, we calculated a mean score for each participant. Age-related differences in all these thought dimensions were assessed using a series of independent samples t-tests with Bonferroni correction.

## fMRI Data acquisition

Whole-brain imaging was performed on a 3T Siemens Magnetom MRI system (Siemens Medical Systems, Erlangen, Germany) using a 16-channel head coil. A structural MP-RAGE was acquired following a standard acquisition protocol (TR = 2350 ms, TE = 2.26 ms, FOV = 256x256, slice thickness = 1mm, voxel size = 1mm isotropic) as reported in previous work [28, 29]. BOLD- sensitive T2*-weighted functional images were acquired using a single shot gradient-echo EPI pulse sequence (TR = 2000ms, TE = 30ms, flip angle = 78˚, 32 axial slices, 3.5×3.5×4.0mm, distance factor 0%, FoV = 192×192mm, interleaved slice ordering) and corrected online for head motion. Head motion was restricted using firm padding that surrounded the head. The first two volumes were discarded to allow for T1 equilibration effects.

## fMRI preprocessing

fMRI data were preprocessed in SPM12. Images were first realigned, and then slice-time corrected. For normalization, we used the DARTEL procedure, which creates a study-specific template. Each participant's anatomical image was co-registered to the mean EPI image generated in the realignment step. The co-registered images were then segmented, and the gray and white matter segmented images were used to create a study-specific template. The flow fields containing the deformation parameters to this template for each subject were used to normalize each participant's realigned and resliced EPIs to MNI space. Normalized EPI images were smoothed with a 6mm full-width half-maximum isotropic Gaussian kernel.

## fMRI data analysis

To determine whether there were age-related differences in movement that could influence the fMRI analysis, we assessed whether there were age-related differences in the mean value of any of the six movement parameters (x, y, z, roll, pitch, yaw) using six independent sample t-tests. The main focus of the fMRI analysis was age-related differences in the neural correlates of stimulus-independent thought versus external focus of attention ("3" and "4" responses versus "2" "1" responses). Statistical analyses were performed in 2 stages of a mixed effects model. At the first level, an 8 second epoch preceding each thought probe was assigned to stimulus-independent thought or external focus categories based on answers to the thought probe and convolved with the canonical hemodynamic response function. The thought probes were modeled as a single epoch beginning at the onset of the first question (asking about stimulus-independent thought versus external focus) and lasting until the offset of the last question (asking about novelty). Serial correlations were accounted for using an autoregressive AR(1) model. A high-pass filter cut-off of 128 was used, and no global normalization was performed. Finally, the 6 movement parameters were included as regressors of no interest.

Two t-test contrasts were specified per subject: stimulus-independent thought versus external focus and external focus versus stimulus-independent thought. These contrasts were carried forward to a second stage in which subjects were treated as a random effect. At the second level, we assessed age-related similarities and differences using independent samples t-tests. To account for age-related differences in content of thought, we entered any variable that exhibited an age-related difference as a covariate in the $2^{nd}$ level model. The results were thesholded at a voxel-wise $p < 0.001$ level with k = 25, cluster-corrected at FWE $p < 0.05$. One concern with this relatively stringent threshold is that it may result in type-2 errors and deficient meta-analyses [30]. Therefore, we also separately report all regions that passed a voxel-wise $p < 0.001$ threshold with no cluster correction but do not discuss the regions any further.

Although our primary interest was in stimulus-independent thought versus external focus of attention, we also collected data regarding temporality, person-centeredness and goal-directedness of thought. These data were collected so that we could determine, at a behavioral level, whether there are age-related differences in the content of thought. Although it would have been of interest to assess age-related differences in these dimensions of thought, such analyses were not well powered because 1) these characteristics of thought were only assessed when participants reported a thought (rather than an external focus), and 2) because this split these events up into further categories (e.g. present, past, future, atemporal), reducing event number. Therefore, we do not discuss these further.

## Results

### Behavioral results

Behavioral results are available at https://osf.io/snmdv/. We did not find age-related differences in any of the six movement parameters (all $p > 0.10$). Behavioral results are presented in Table 1. The mean score for the stimulus-independent score (ranging from 1 to 4) did not

**Table 1. Behavioural results, mean (SD).**

|  | Young | Old | Significance |
|---|---|---|---|
| *Stimulus-independence* |  |  |  |
| 1 (Fully external) | 0.11 (0.09 | 0.19 (0.17) |  |
| 2 (Mostly external) | 0.29 (0.17) | 0.24 (0.15) |  |
| 3 (Mostly Internal) | 0.36 (0.16) | 0.32 (0.19) |  |
| 4 (Fully Internal) | 0.24 (0.22) | 0.25 (0.18) |  |
| Mean score | 2.73 (0.39) | 2.62 (0.49) | $t(49) = 0.85, p = 0.40, d = 0.24$ |
| *Temporality* |  |  |  |
| Past | 0.21 (0.13) | 0.14 (0.16) | $t(49) = 1.63, p = 0.11, d = 0.46$ |
| Present | 0.31 (0.19) | 0.52 (0.20) | $t(49) = 3.75, p < 0.001, d = 1.06$ |
| Future | 0.29 (0.16) | 0.30 (0.17) | $t(49) = 0.19, p = 0.85, d = 0.05$ |
| Atemporal | 0.18 (0.15) | 0.04 (0.06) | $U = 145, p < 0.001, d = 1.23$ |
| *Goal-directedness* |  |  |  |
| Goal-directed | 0.65 (0.21) | 0.52 (0.26) | $t(49) = 1.86, p = 0.07, d = 0.53$ |
| Not goal-directed | 0.34 (0.21) | 0.48 (0.26) |  |
| *Person-Centeredness* |  |  |  |
| Self only | 0.40 (0.14) | 0.50 (0.27) | $t(49) = 1.77, p = 0.08, d = 0.50$ |
| Others only | 0.14 (0.07) | 0.13 (0.11) | $t(49) = 0.22, p = 0.83, d = 0.06$ |
| Self and others | 0.36 (0.14) | 0.32 (0.22) | $t(49) = 0.77, p = 0.44, d = 0.22$ |
| None | 0.11 (0.08) | 0.05 (0.07) | $t(49) = 2.69, p = 0.01, d = 0.76$ |
| *Mean novelty* | 2.29 (0.57) | 2.69 (0.57 | $t(49) = 2.52, p = 0.02, d = 0.72$ |

differ between young and older adults, $t(49) = 0.85$, $p = 0.40$, $d = 0.24$. Note that there also was no age-related difference in proportion of any of the four levels of stimulus-independence after correction for four comparisons (all p > 0.0125).

For temporality, we used a threshold of p < 0.0125 to correct for four comparisons. Young adults had more atemporal thoughts than older adults U = 145, $p < 0.001$, $d = 1.23$, who instead reported more present-oriented thoughts $t(49) = 3.75$, $p < 0.001$, $d = 1.06$. Note that a Mann-Whitney U test was used for atemporal thoughts because of a violation of the assumption of equal variances according to Levene's test (p< 0.05). However, a similar result (more atemporal thoughts in young versus older adults) would have been obtained using a t-test (p < 0.001). There were no age-related differences in past-oriented thoughts, $t(49) = 1.63$, $p = 0.11$, $d = 0.46$, or future-oriented thoughts, $t(49) = 0.19$, $p = 0.85$, $d = 0.05$.

For person-centeredness, we used a threshold of p < 0.0125 to correct for four comparisons. Young adults chose the "none" option to a greater extent than old, $t(49) = 2.69$, $p = 0.01$, $d = 0.76$, and there was a trend for older adults to have more self-focused thoughts compared to young, $t(49) = 1.77$, $p = 0.08$, $d = 0.5$. There were no age-related differences in thoughts about others, $t(49) = 0.22$, $p = 0.83$, $d = 0.06$, or in thoughts and the self and others, $t(49) = 0.77$, $p = 0.44$, $d = 0.22$.

Older adults reported that their thoughts were more novel than did younger participants $t(49) = 2.52$, $p = 0.02$, $d = 0.72$. Finally, there was a trend for young adults to have more goal-directed thoughts compared to old, $t(49) = 1.86$, $p = 0.07$, $d = 0.53$.

## fMRI results

fMRI results controlled for all variables exhibiting a significant age difference behaviorally (% of atemporal thoughts, present-oriented thoughts, mean novelty of thought, the amount of "none" responses selected for person-centeredness) and in those showing trends for age differences (goal-directedness and self-related thoughts). Across groups, there was greater activation in posterior cingulate and angular gyrus for stimulus-independent thought vs. external focus (Fig 1; Table 2). In contrast, no regions survived FWE correction for external focus vs. stimulus-independent thought across groups. We observed an age by thought interaction in medial and left lateral prefrontal cortex as well as left superior temporal gyrus: there was greater recruitment of these regions in young versus older adults for stimulus-independent thought vs. external focus. No other significant effects were observed.

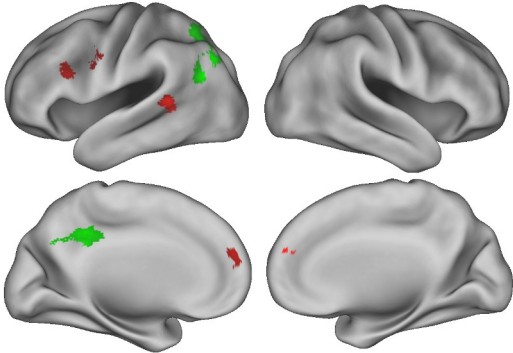

**Fig 1. This figure depicts the fMRI results for the stimulus-independent thought versus external focus contrast with FWE cluster-wise correction.** Regions colored in green were activated across age groups during stimulus-independent thought versus external attention whereas young adults had greater recruitment of red regions compared to older adults.

**Table 2. fMRI activation results.**

| | MNI coordinates | Cluster extent | Voxel Peak t | Cluster p FWE-corrected |
|---|---|---|---|---|
| **Group similarities, Stimulus-independent thought > external focus** | | | | |
| *Voxel-level, p < 0.001, Cluster-level, FWE p < 0.05, k >25* | | | | |
| Left Angular Gyrus | -30 -63 46 | 1286 | 5.5 | <0.001 |
| | -46 -69 34 | | 5.1 | |
| | -44 -63 26 | | 4.82 | |
| Left Precuneus | -4 -45 34 | 457 | 4.62 | 0.014 |
| | -2 -64 32 | | 4.45 | |
| | -6 -57 33 | | 3.85 | |
| *Voxel level, p < 0.001, Cluster level, no cluster-level correction, k > 25* | | | | |
| Left Postcentral Gyrus / inferior parietal lobe | -40 -28 58 | 73 | 5.59 | 0.803 |
| | -30 -28 68 | | 5.19 | |
| | -33 -32 56 | | 4.61 | |
| Left Precentral Gyrus | -30 -18 62 | 195 | 5.59 | 0.227 |
| | -38 -15 58 | | 4.97 | |
| | -28 -9 57 | | 4.03 | |
| Left Medial Frontal Gyrus | -6 6 62 | 55 | 5.35 | 0.898 |
| Left Precentral Gyrus | -27 -15 70 | 31 | 5.28 | 0.979 |
| Right Cerebellum | 21 -46 -26 | 60 | 4.46 | 0.874 |
| Right Cerebellum | 28 -63 -30 | 125 | 4.41 | 0.497 |
| | 42 -58 -28 | | 3.82 | |
| Right Precentral Gyrus | 39 -6 57 | 82 | 4.39 | 0.749 |
| | 32 -12 62 | | 3.46 | |
| Right Cerebellum | 4 -48 -38 | 106 | 4.27 | 0.603 |
| | 8 -42 -46 | | 3.46 | |
| Right Paracentral Lobule | 12 -32 72 | 34 | 4.02 | 0.972 |
| **Group similarities, External focus > Stimulus-independent thought** | | | | |
| *Voxel-level, p < 0.001, Cluster-level, FWE p < 0.05, k >25* | | | | |
| No significant clusters | | | | |
| *Voxel level, p < 0.001, Cluster level, no cluster-level correction, k > 25* | | | | |
| Left Cerebellum | -14 -58 -3 | 283 | 5.39 | 0.084 |
| Right Superior Frontal Gyrus / frontal pole | 20 48 28 | 91 | 4.53 | 0.694 |
| Left Parahippocampal Gyrus | -34 -48 -6 | 67 | 4.41 | 0.837 |
| Left Supramarginal Gyrus/ | -62 -38 36 | 59 | 4.25 | 0.879 |
| Inferior Parietal Lobule | -62 -27 33 | | 4.01 | |
| Right Inferior Frontal Gyrus / frontal pole | 39 39 10 | 184 | 4.08 | 0.257 |
| | 38 27 14 | | 3.81 | |
| Left Cerebellum | -28 -56 -40 | 31 | 3.97 | 0.979 |
| Left Caudate | -21 4 30 | 123 | 3.89 | 0.508 |
| Right Inferior Parietal Lobule | 60 -28 36 | 29 | 3.88 | 0.983 |
| Right Inferior Frontal Gyrus | 52 15 9 | 38 | 3.8 | 0.962 |
| Right Superior Frontal Gyrus | 8 36 48 | 28 | 3.79 | 0.984 |
| | 34 21 21 | 45 | 3.77 | 0.94 |
| Right Inferior Frontal Gyrus | 45 40 -3 | 27 | 3.73 | 0.986 |
| **Group Interaction: Young > Old, Stimulus-independent thought > External focus** | | | | |
| *Voxel-level, p < 0.001, Cluster-level, FWE p < 0.05, k >25* | | | | |
| Left Superior Temporal Gyrus / supramarginal gyrus | -56 -42 9 | 562 | 6.55 | 0.005 |
| | -46 -48 2 | | 4.63 | |

*(Continued)*

**Table 2.** (Continued)

| | MNI coordinates | Cluster extent | Voxel Peak t | Cluster p FWE-corrected |
|---|---|---|---|---|
| Left precentral gyrus / Inferior Frontal Gyrus | -52 8 32 | 604 | 4.66 | 0.003 |
| Left Inferior Frontal Gyrus | -48 16 22 | | 4.57 | |
| Left Precentral Gyrus / Middle Frontal Gyrus | -46 4 40 | | 3.9 | |
| Left Medial Prefrontal Cortex | -2 48 21 | 396 | 4.47 | 0.025 |
| | -8 56 21 | | 4.23 | |
| | 8 54 20 | | 3.57 | |
| *Voxel level, p < 0.001, Cluster level, no cluster-level correction, k > 25* | | | | |
| Left Supplementary Motor Cortex | -3 4 50 | 165 | 4.79 | 0.32 |
| Left Temporal Pole | -50 12 -12 | 246 | 4.72 | 0.127 |
| Left Insula | -45 8 -3 | | 4.24 | |
| Right Middle/Superior Temporal Gyrus | 57 -8 -12 | 161 | 4.67 | 0.334 |
| | 57 3 -16 | | 3.98 | |
| Right Cerebellum | 0 -68 -30 | 120 | 4.49 | 0.524 |
| Left Cingulate Gyrus | -2 9 38 | 93 | 4.4 | 0.682 |
| Left Cingulate Gyrus | -8 18 34 | | 3.4 | |
| Left Superior Temporal Gyrus / Supramarginal Gyrus | -58 -48 20 | 55 | 4.37 | 0.898 |
| Left Precentral Gyrus | -34 -22 57 | 140 | 4.33 | 0.423 |
| Left Postcentral Gyrus | -42 -21 52 | | 3.75 | |
| Cingulate gyrus | 2 -20 46 | 271 | 4.31 | 0.095 |
| Left Supplementary Motor Cortex | -8 -9 48 | | 3.9 | |
| Left Insula | -34 -18 16 | 94 | 4.29 | 0.676 |
| Right Medial Prefrontal Cortex | 9 54 9 | 110 | 4.27 | 0.58 |
| Right Frontal Pole | 21 62 12 | | 3.88 | |
| Left Parietal Operculum | -57 -26 15 | 264 | 4.17 | 0.103 |
| Left Heschl's gyrus | -48 -21 12 | | 4.02 | |
| Left Opercular Cortex | -60 -18 14 | | 3.75 | |
| Left Postcentral Gyrus | -45 -22 42 | 27 | 4.12 | 0.986 |
| Right Postcentral Gyrus | 3 -33 57 | 78 | 4.12 | 0.773 |
| Left Anterior Cingulate | -4 48 6 | 53 | 4.03 | 0.907 |
| Left Parahippocampal Gyrus | -6 -40 3 | 42 | 3.98 | 0.95 |
| Right Opercular Cortex | 46 2 8 | 44 | 3.93 | 0.943 |
| Right Posterior Cingulate | 4 -58 20 | 84 | 3.87 | 0.737 |
| Left Middle Temporal Gyrus | -44 -33 -3 | 27 | 3.87 | 0.986 |
| | -45 -24 -8 | | 3.37 | |
| Left Medial Prefrontal Gyrus | -3 50 30 | 44 | 3.84 | 0.943 |
| Right Inferior Frontal Gyrus | 36 38 -12 | 31 | 3.72 | 0.979 |
| Right Cingulate Gyrus | 3 16 24 | 51 | 3.72 | 0.916 |
| Left Superior Temporal Gyrus | -58 -8 0 | 45 | 3.72 | 0.94 |
| Left Precentral Gyrus | -32 -2 52 | 38 | 3.72 | 0.962 |
| Right Superior Frontal Gyrus | 24 33 45 | 35 | 3.71 | 0.97 |
| Left Superior Temporal Gyrus | -52 -3 -16 | 50 | 3.71 | 0.92 |
| Left Precentral Gyrus | -6 -20 58 | 27 | 3.57 | 0.986 |
| Left Cingulate Gyrus | 0 -10 32 | 28 | 3.5 | 0.984 |

Note: This table presents the between-group fMRI results for the stimulus-independent thought > external focus of attention contrast. The table presents results that passed a threshold of p < 0.001 voxel-level with k = 25, cluster-corrected at FWE p < 0.05, as well as those regions that passed a p < 0.001 voxel-level with k = 25 without cluster correction.

## Discussion

In recent years, many studies have reported that healthy older adults spend less time engaged in task-unrelated thoughts compared to young adults [1, 2]. In contrast, much less is known regarding age-related differences in stimulus-independent thoughts in the absence of an ongoing task. The goal of the current study was to assess age-related differences in time spent engaging in stimulus-independent thoughts, their content and their neural correlates in the absence of an ongoing task.

In contrast with age-related reductions in task-unrelated thoughts [1], we found that younger and older adults spent a similar amount of time engaged in stimulus-independent thought. Age-related reductions during experimental tasks have been attributed to factors including reduced cognitive resources and increased task interest/motivation with age. These factors were minimized in the current paradigm which did not include an ongoing task, providing a possible explanation for the lack of age differences in the current study (though null results, particularly with small sample sizes, should be interpreted with caution). Our findings support suggestions that minimally demanding cognitive tasks may be ideal to promote high levels of stimulus-independent thoughts in older adults [2, 4, 8, 18].

Next, we turned to the content of stimulus-independent thought. Older adults reported that their thoughts were more present-oriented, novel and asocial compared to young adults. Although speculative, one possibility is that these differences arose because older adults had more thoughts about the scanner environment compared to young adults (e.g. wondering how the MRI scanner works). These thoughts may have been rated as novel because people do not typically think about MRI environments, present-oriented because they were in an MRI environment, and asocial due the thoughts not being about anybody in particular. This possibility is consistent with suggestions that older adults and dementia patients may have increasing difficulty exhibiting thoughts completely detached from ongoing events [8, 31]. Rather than using thought probes with multiple choice questions as we did here, future studies could benefit from using open-ended thought probes allowing participants to verbally describe their thoughts [32]. Such thought probes may allow a more fine-grained analysis of age-related differences in content of thought than is possible here. We were unable to examine neural correlates associated with these age-related differences in content of thought because of a lack of events. Studying unconstrained thought comes with the problem that it is hard to predict what participants will think about. Thus, longer scanning sessions than the one used here should be used if the goal is to compare different thought types.

Although we were unable to assess the neural correlates of specific types of stimulus-independent thoughts, we were able to assess age-related differences in stimulus-independent thoughts considered as a whole versus external focus of attention. Younger and older adults activated the posterior cingulate and left angular gyrus, two regions commonly implicated in mind-wandering and related forms of spontaneous thought [13], as well as memory retrieval [33], future thinking [34], and self- or other-directed thoughts [35]. It has been proposed that these regions may be involved in stimulus-independent thoughts containing a high amount of detail [36].

We also observed age-related differences in the neural correlates of stimulus-independent thoughts. Specifically, compared to older adults, young adults displayed greater activation in several regions including medial prefrontal cortex, left lateral prefrontal cortex, and left superior temporal gyrus, all of which have previously been implicated in mind-wandering in young adults [e.g., 12]. This observation goes somewhat against the more common orthodoxy of *increased* age-related recruitment of lateral prefrontal cortex during a variety of cognitive tasks [37]. However, it is consistent with prior studies of task-unrelated thought which have

implicated these regions in age-related differences in ongoing thoughts with age [16, 18]. For instance, Martinon, Riby [16] found that altered connectivity between lateral temporal cortex and both ventrolateral and dorsomedial cortex was associated with a reduced ability to up-regulate patterns of ongoing thoughts in older adults. In the current study, although young and older adults spent a similar amount of time engaged in stimulus-independent thoughts, it is possible that young adults exhibited greater control over their streams of thought, or that these thoughts were more dominated by planning and deliberation in young versus older adults. While it is true that the fMRI model controlled for age-related differences in content, including goal-relatedness, which could mitigate this possibility, it is still possible that other aspects of thoughts that we did not measure, such as intentionality [38], account for these differences.

In summary, in the absence of an ongoing task, we found that whereas young and older adults spent a similar amount of time engaged in stimulus-independent thoughts, their content and neural correlates differed across the adult lifespan. Our findings suggest that the absence of an ongoing task may constitute a state wherein differences in cognition due to age are likely to occur.

## Appendix - Task instructions

General Instructions: For this task, you will be asked to simply relax in the scanner and answer questions about your thoughts at different points in time. Try to be as natural as possible and just think as you would normally. We're not looking for one type of thought or another. Try to answer the questions as accurately as you can.

Thought Question 1: Every 8–18 seconds, you will be asked whether you were JUST having any thoughts (i.e., thoughts right before you see the question)

1 = Completed focused externally

Directly related to the environment/no thoughts

Visual, auditory, body sensations, scanner sounds, etc.

Trial ends if you answer this

2 = Somewhat externally focused

Partially focused on the environment

3 = Somewhat focused on my thoughts

Thoughts, memories, plans for the future, daydreams, etc.

Most recent thought

E.g., what you will do later, what you did last night, or whatever comes to mind

4 = Completely focused on my thoughts

Immersed in what you are thinking about

Thought Question 2: Where were these thoughts focused in terms of time?

1 = Past; 2 = Present; 3 = Future; 4 = None

Thought Question 3: Were the thoughts related or unrelated to your current concerns or goals?

Your personal concerns or goals in life right now (but not about the experiment)

Doesn't matter if other people would have the same goals or not

1 = Related

2 = Unrelated

Thought Question 4: Did the thoughts involve you and/or other people?

1 = Self only

2 = Others only

3 = Self + Others

4 = None

Thought Question 5: How similar or different were the thoughts you were just having to your previous thoughts?

Some thoughts are common and similar to what you might have thought about before the experiment (e.g., thinking about dinner plans)

Some thoughts are more uncommon and different from what you thought about before the experiment (e.g., wondering how the MRI works)

Both types of thoughts are important

1 = very similar

2 = somewhat similar

3 = somewhat different

4 = very different

## Author Contributions

**Conceptualization:** Roger E. Beaty, Kieran C. R. Fox, R. Nathan Spreng.

**Formal analysis:** David Maillet.

**Funding acquisition:** Roger E. Beaty.

**Investigation:** Roger E. Beaty.

**Methodology:** Roger E. Beaty.

**Project administration:** Roger E. Beaty, R. Nathan Spreng.

**Supervision:** R. Nathan Spreng.

**Visualization:** David Maillet.

**Writing – original draft:** David Maillet, Areeba Adnan, Kieran C. R. Fox, Gary R. Turner, R. Nathan Spreng.

**Writing – review & editing:** David Maillet, Roger E. Beaty, Areeba Adnan, Kieran C. R. Fox, Gary R. Turner, R. Nathan Spreng.

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
