## [Decision Letter · Decision Letter 0]

18 Sep 2019

PONE-D-19-23089

Aging and the wandering brain: Age-related differences in the neural correlates of stimulus-independent thoughts

PLOS ONE

Dear Dr. Maillet,

Thank you for submitting your manuscript to PLOS ONE. After careful consideration by two Reviewers and an Academic Editor, please make the suggested corrections posed by the Reviewers so I can render a decision on this manuscript.

**Comments to the Author**

1. Is the manuscript technically sound, and do the data support the conclusions?

Reviewer #1: Yes

Reviewer #2: Yes

2. Has the statistical analysis been performed appropriately and rigorously? 

Reviewer #1: Yes

Reviewer #2: Yes

3. Have the authors made all data underlying the findings in their manuscript fully available?

Reviewer #1: Yes

Reviewer #2: Yes

4. Is the manuscript presented in an intelligible fashion and written in standard English?

Reviewer #1: Yes

Reviewer #2: Yes

5. Review Comments to the Author

Reviewer #1:

1. There is an extra bracket in a citation on page 5 of the manuscript: “(Mevel et al., (2013).”

2. It appears as though Mann-Whitney U tests were used to examine pair-wise differences in atemporal thought between young and older adults, but this decision is neither justified nor self-explanatory. If there were differences in the underlying data that caused the authors to use different statistics even within the same family of tests (i.e., temporality tests), this should at least be addressed in the manuscript in some way.

3. The authors justify the originality of their study by stating that while similar studies have been done in the past examining task-unrelated thought, this is the first of its kind to study stimulus-independent thought (in addition to the taskless paradigm). It is largely left to the reader to infer why this is important to study; very little explanation of what stimulus-independent thought is has been offered either in the introduction or the discussion. The authors do not discuss either its real-world significance or its similarities and differences to task-unrelated thought with which it is being implicitly contrasted. The results are certainly interesting and meaningful, but they are given virtually free of context. While I understand the desire to keep manuscripts as short as possible and eliminate unnecessary details, I believe a small amount of background and context regarding the variables of interest and why they have been chosen would benefit the manuscript overall.

Reviewer #2: This is an interesting study which is well structed and clearly explained. The study revealed that both age groups reported a similar amount of time engaged in stimulus-independent thoughts, older adults rated their thoughts as more present-oriented and more novel. Across age groups the experiencing of stimulus-independent thoughts was associated with increased posterior cingulate and left angular gyrus activation compared to an external focus of attention. When experiencing stimulus-independent thoughts, younger adults engaged medial and left lateral prefrontal cortex as well as left superior temporal gyrus to a greater degree than older adults.

I have a few recommendations.

The experimental design written in the section 2.2. procedure is not entirely clear. How long did rest last per run? The authors wrote «The thought probes appeared after 8, 10, 12, 14, 16, or 18 seconds (6 of each per run)». Were there only 6 «rests» per run? Were 8, 10, 12, 14, 16 and 18 seconds chosen at random? How long did the experiment last at average?

In the discussion section, it is advisable to discuss the result which indicated that older adults describe their stimulus independent thoughts as more novel and focused on the present.

6. PLOS authors have the option to publish the peer review history of their article (what does this mean?). If published, this will include your full peer review and any attached files.

**Do you want your identity to be public for this peer review?** For information about this choice, including consent withdrawal, please see our Privacy Policy.

Reviewer #1: No

Reviewer #2: No

We would appreciate receiving your revised manuscript by December, 2020. To enhance the reproducibility of your results, we recommend that if applicable you deposit your laboratory protocols in protocols.io, where a protocol can be assigned its own identifier (DOI) such that it can be cited independently in the future. For instructions see: http://journals.plos.org/plosone/s/submission-guidelines#loc-laboratory-protocols

We look forward to receiving your revised manuscript.

Kind regards,

Stephen D. Ginsberg, Ph.D.

Section Editor

PLOS ONE

1 Please ensure that your manuscript meets PLOS ONE's style requirements, including those for file naming. The PLOS ONE style templates can be found at

---

## [Author Response · Author response to Decision Letter 0]

27 Sep 2019

Response to reviewers

We thank the reviewers for their helpful feedback, which we believe have helped to considerably improve the quality of the manuscript. Below, we address each comment in turn.

Reviewer #1:

1. There is an extra bracket in a citation on page 5 of the manuscript: “(Mevel et al., (2013).”

Response: Thank you for pointing this out, it has been corrected.

2. It appears as though Mann-Whitney U tests were used to examine pair-wise differences in atemporal thought between young and older adults, but this decision is neither justified nor self-explanatory. If there were differences in the underlying data that caused the authors to use different statistics even within the same family of tests (i.e., temporality tests), this should at least be addressed in the manuscript in some way.

Response: We apologize for not properly justifying the use of this test. On pages 14-15, we now state that: Young adults had more atemporal thoughts than older adults U = 145, p < 0.001, d = 1.23, who instead reported more present-oriented thoughts t(49) = 3.75, p < 0.001, d = 1.06. Note that a Mann-Whitney U test was used for atemporal thoughts because of a violation of the assumption of equal variances according to Levene’s test (p< 0.05). However, a similar result (more atemporal thoughts in young versus older adults) would have been obtained using a t-test (p < 0.001).

3. The authors justify the originality of their study by stating that while similar studies have been done in the past examining task-unrelated thought, this is the first of its kind to study stimulus-independent thought (in addition to the taskless paradigm). It is largely left to the reader to infer why this is important to study; very little explanation of what stimulus-independent thought is has been offered either in the introduction or the discussion. The authors do not discuss either its real-world significance or its similarities and differences to task-unrelated thought with which it is being implicitly contrasted. The results are certainly interesting and meaningful, but they are given virtually free of context. While I understand the desire to keep manuscripts as short as possible and eliminate unnecessary details, I believe a small amount of background and context regarding the variables of interest and why they have been chosen would benefit the manuscript overall.

Response: We thank the reviewer for pointing this out. We believe one of the most interesting reasons to study stimulus-independent thoughts in the absence of a task is to assess whether findings of age-related reductions in time spent engaged in task-unrelated thoughts extend to this type of paradigm. Specifically, on page 5, we state that:

This paradigm was of interested for several reasons. First, many of the factors that have been used to explain age-related reductions in task-unrelated thoughts, such as reductions in cognitive resources, and increased motivation to perform ongoing tasks should be minimized in the absence of a task. These accounts predict that young and older adults should spend a similar amount of time engaged in stimulus-independent thoughts in the absence on an ongoing task, a prediction that we tested here.

Second, regarding the specific characteristics of thoughts chosen, on pages 6-7, we now state (novel text is underlined): 

Second, the experience of letting one’s thoughts wander in the absence of any external task occurs frequently in daily life (e.g. while sitting on a bus, or on a chair by the lake) but we know next to nothing about differences in what young and older adults may be thinking about during these times. In the current experiment, we were interested in age-related differences in several characteristics of stimulus-independent thoughts including temporality, novelty, goal-relatedness, and person-centeredness (self, other, both, or none). We chose these characteristics because although much prior aging research has focused on the detrimental aspects of mind-wandering, such as its link to inattention (Maillet & Rajah, 2016; McVay et al., 2013) and negative mood (Frank et al., 2015; Maillet et al., 2018), there is reason to believe that these thoughts may also confer certain benefits. In particular, whereas there now exists considerable evidence of a link between mind-wandering and creativity in young adults (for a review, see K. C. Fox & Beaty, 2019), we do not know of any prior research that has assessed the creativity/novelty of thoughts in older adults. There is also evidence that a large proportion on stimulus-independent thoughts in young adults are social (K. C. R. Fox et al., 2018) and goal-directed (Ralph, Wammes, Barr, & Smilek, 2017; Stawarczyk, Cassol, & D'Argembeau, 2013) in nature, yet relatively little data is available in older adults.

A characteristic of thought that has often been studied in the aging and mind-wandering literature and that we also measured here is temporality of thought. Whereas some studies have failed to observe age-related differences in temporality of thought (Maillet et al., 2018; Maillet & Schacter, 2016b), another found that young adults reported more future-oriented mind-wandering compared to older adults who instead reported more past-oriented mind-wandering (Irish, Goldberg, Alaeddin, O'Callaghan, & Andrews-Hanna, 2019). It was suggested that young adults might engage in more simulation of future outcomes, planning, and decision-making compared to older adults, who might instead interpret, make sense of, and possibly derive satisfaction from remembrance of the past. In that study, older adults also exhibited fewer self-related thoughts compared to young, which the authors suggested could help protect older adults from negative mood and contribute to the age-related positivity bias (Carstensen, 1992). Thus, in the current study, one possibility is that young adults would report more future-oriented and self-related thoughts compared to old, indicative of a greater prospective bias in which young adults plan for future outcomes involving themselves. Because future-oriented thoughts tend to be more novel and goal-oriented (Cole & Berntsen, 2016), young adults might also exhibit more of these types of thoughts compared to older adults.

Reviewer #2: This is an interesting study which is well structed and clearly explained. The study revealed that both age groups reported a similar amount of time engaged in stimulus-independent thoughts, older adults rated their thoughts as more present-oriented and more novel. Across age groups the experiencing of stimulus-independent thoughts was associated with increased posterior cingulate and left angular gyrus activation compared to an external focus of attention. When experiencing stimulus-independent thoughts, younger adults engaged medial and left lateral prefrontal cortex as well as left superior temporal gyrus to a greater degree than older adults. I have a few recommendations.

The experimental design written in the section 2.2. procedure is not entirely clear. How long did rest last per run? The authors wrote «The thought probes appeared after 8, 10, 12, 14, 16, or 18 seconds (6 of each per run)». Were there only 6 «rests» per run? Were 8, 10, 12, 14, 16 and 18 seconds chosen at random? How long did the experiment last at average?

Response: We apologize for the confusion. On Page 9, we now state:

In a single scanning session, participants completed two fMRI runs that alternated between periods of “rest”, in which there was no explicit task, and thought probes, that asked participants about the content of their thoughts in the directly preceding moment. Each run contained 48 thought probes (total of 96 overall). The thought probes appeared after 8, 10, 12, 14, 16, or 18 seconds with equal probability throughout the experiment and in a random order. Answers to all thought probe questions were self-paced. Thus, duration of each scanning run differed for every subject depending on how long they took to answer the probe questions. On average, each run lasted 15.5 minutes (SD = 1.98) in young adults and 17.6 minutes (SD = 3.38) in older adults, t(1,49) = 2.75, p < 0.01.

In the discussion section, it is advisable to discuss the result which indicated that older adults describe their stimulus independent thoughts as more novel and focused on the present.

On page 21, we now state that (novel text is underlined): Next, we turned to the content of stimulus-independent thought. Older adults reported that their thoughts were more present-oriented, novel and asocial compared to young adults. Although speculative, one possibility is that these differences arose because older adults had more thoughts about the scanner environment compared to young adults (e.g. wondering how the MRI scanner works). These thoughts may have been rated as novel because people do not typically think about MRI environments, present-oriented because they were in an MRI environment, and asocial due the thoughts not being about anybody in particular. This possibility is consistent with suggestions that older adults and dementia patients may have increasing difficulty exhibiting thoughts completely detached from ongoing events (Maillet & Schacter, 2016b; O'Callaghan, Shine, Hodges, Andrews-Hanna, & Irish, 2019). Rather than using thought probes with multiple choice questions as we did here, future studies could benefit from using open-ended thought probes allowing participants to verbally describe their thoughts (Jordao, Pinho, & St Jacques, 2019). Such thought probes may allow a more fine-grained analysis of age-related differences in content of thought than is possible here. We were unable to examine neural correlates associated with these age-related differences in content of thought because of a lack of events. Studying unconstrained thought comes with the problem that it is hard to predict what participants will think about. Thus, longer scanning sessions than the one used here should be used if the goal is to compare different thought types.

---

## [Editor Report · Decision Letter 1]

3 Oct 2019

Aging and the wandering brain: Age-related differences in the neural correlates of stimulus-independent thoughts

PONE-D-19-23089R1

Dear Dr. Maillet,

We are pleased to inform you that your manuscript has been judged scientifically suitable for publication and will be formally accepted for publication once it complies with all outstanding technical requirements.

With kind regards,

Stephen D. Ginsberg, Ph.D.

Section Editor

PLOS ONE

---

## [Editor Report · Acceptance letter]

7 Oct 2019

PONE-D-19-23089R1 

Aging and the wandering brain: Age-related differences in the neural correlates of stimulus-independent thoughts 

Dear Dr. Maillet:

I am pleased to inform you that your manuscript has been deemed suitable for publication in PLOS ONE. Congratulations! Your manuscript is now with our production department. 

With kind regards,

on behalf of

Dr. Stephen D Ginsberg 

Section Editor

PLOS ONE